# Gender Specific Differences in Disease Susceptibility: The Role of Epigenetics

**DOI:** 10.3390/biomedicines9060652

**Published:** 2021-06-08

**Authors:** Lucia Migliore, Vanessa Nicolì, Andrea Stoccoro

**Affiliations:** 1Department of Translational Research and of New Surgical and Medical Technologies, Medical School, University of Pisa, 56126 Pisa, Italy; vanessa.nicoli@med.unipi.it (V.N.); andrea.stoccoro@unipi.it (A.S.); 2Department of Laboratory Medicine, Azienda Ospedaliero Universitaria Pisana, 56124 Pisa, Italy

**Keywords:** epigenetics, gender bias, complex diseases, infectious diseases, vaccination outcomes, skewed X inactivation, escape genes, SARS-CoV-2 infection

## Abstract

Many complex traits or diseases, such as infectious and autoimmune diseases, cancer, xenobiotics exposure, neurodevelopmental and neurodegenerative diseases, as well as the outcome of vaccination, show a differential susceptibility between males and females. In general, the female immune system responds more efficiently to pathogens. However, this can lead to over-reactive immune responses, which may explain the higher presence of autoimmune diseases in women, but also potentially the more adverse effects of vaccination in females compared with in males. Many clinical and epidemiological studies reported, for the SARS-CoV-2 infection, a gender-biased differential response; however, the majority of reports dealt with a comparable morbidity, with males, however, showing higher COVID-19 adverse outcomes. Although gender differences in immune responses have been studied predominantly within the context of sex hormone effects, some other mechanisms have been invoked: cellular mosaicism, skewed X chromosome inactivation, genes escaping X chromosome inactivation, and miRNAs encoded on the X chromosome. The hormonal hypothesis as well as other mechanisms will be examined and discussed in the light of the most recent epigenetic findings in the field, as the concept that epigenetics is the unifying mechanism in explaining gender-specific differences is increasingly emerging.

## 1. Introduction

In humans, many complex traits or diseases have an altered sex ratio, where, by complex trait/disease, we mean a phenotype that is typically due to the interaction between genetic and environmental factors.

In determining gender, many factors are involved, such as culture, ethnicity, socio economic background, religion, and many others [1], not only the composition of gonosomes (determining normally the genetic sex, with the exception of disorders of sexual development). Consequently, gender can be considered an environmental factor (intrinsic) of susceptibility as well as age or environmental exposures, able to interfere with the risk of many complex diseases. Moreover, it is well known that environmental factors, which play an important role in modulating the susceptibility to diseases in men and women, can induce epigenetic modifications. For the above reasons we will refer to gender-specific susceptibility to complex traits/diseases.

A differential susceptibility of males and females to autoimmune diseases, some cancers, infectious diseases, toxic compounds exposure, as well as a gender difference in affecting the outcome of vaccination has been reported (see Figure 1). A recent literature finding concerns the different susceptibility to SARS-CoV-2 infection, with men with COVID-19 being more at risk for worse outcomes and death [2].

Several hypotheses have been made regarding gender-specific differences in disease susceptibility that attempt to explain fundamental aspects that fall within the sphere of gender medicine. We will examine and discuss these in the light of epigenetic mechanisms.

Epigenetic mechanisms finely regulate the levels of gene expression and play a fundamental role in embryonic development, differentiation, and the maintenance of cellular identity, as well as in many other physiological processes. Only recently, it was realized that, in many cases, these are plastic and dynamic processes in response to environmental factors, and their alteration can contribute to the onset of numerous human diseases. The epigenome is heavily influenced by environmental factors, such as nutrition, chemical pollutants, early traumatic experiences, and changes in temperature and exercise [3].

It is important to underline that the effect of the environment on the epigenome does not concern only the period after birth but is able to influence in an incisive way also the development in utero. Alterations of these epigenetic mechanisms and of their molecular apparatus can have disastrous consequences during the cell differentiation that occurs in the prenatal period and can lead to serious malformations. The work of David Barker, an English pediatrician and epidemiologist, in the 1980s and 1990s, clearly identified the period from conception to the age of 2 (the first 1000 days) as one of the main windows of susceptibility for the health of a person during the course of life [4].

Among complex diseases, tumors undoubtedly constitute the group of pathologies where the role of epigenetics has been amply demonstrated. However, an increasing number of studies suggest that epigenetic modifications also contribute to the development of most of the complex diseases with adult onset, including behavioral pathologies, neurodevelopmental disorders, and autoimmune, metabolic, vascular, and neurodegenerative diseases.

## 2. Diseases with Altered Sex Ratio

A differential susceptibility of males and females in developing autoimmune diseases, cancers, and infectious diseases, as well as regarding a different outcome of vaccination has been extensively documented (Table 1).

Research estimated that 78% of all patients diagnosed with autoimmune pathologies are women [5].

Sjogren’s syndrome (SS), systemic lupus erythematosus (SLE), Grave’s disease (GD), Hashimoto’s autoimmune thyroiditis, and scleroderma show the strongest alteration in sex ratio, with females being seven to ten times more affected than males. The gender bias in autoimmune diseases was also investigated in rheumatoid arthritis (RA), multiple sclerosis (MS), and myasthenia gravis (MG), revealing that the female to male ratio is two to three times more common (2:1–3:1) [6,7,16].

Other immunological disorders that display a sexual dimorphism—which, however, changes through life—are asthma and allergic disease. Asthma is more common in males from birth until puberty but becomes more prevalent and more severe in females after puberty [17]. Furthermore, other diseases where the immune system does not seem involved show differences between males and females: cancer, neurodevelopmental diseases, neurodegenerative, and cardiovascular diseases [8,18].

Cancer development has been observed more in men than women, especially for non-reproductive cancers (with few exceptions, such as thyroid cancer and lung cancer in nonsmokers) or cancers developed in the larynx, esophagus, and bladder, which show a male preponderance of 2:1 and 4:1 respectively, generally with a poor prognosis [18,19,20].

Moreover, many neurodevelopmental disorders with early onset during infancy are characterized by a male preponderance. Autism spectrum disorders (ASD), attention-deficit hyperactivity disorder (ADHD), Tourette’s syndrome, and dyslexia are more prevalent in males, whereas mood disorders, such as phobias, obsessive compulsive disorder, and eating disorders, are more prevalent in females [21,22]. Accumulating evidence on neurodegenerative diseases indicates a gender-dependent bias, with a prevalence of males to females for the majority as well as a severity of the disease differing significantly between the two sexes [23,24].

The differences in the gender ratio observed for many diseases have been the object of many investigations and the roots have been traced back to the interaction of sex chromosomes and hormones. In particular, several mechanisms have been invoked that are peculiarities of the X chromosome, including X-chromosome inactivation, cellular mosaicism, or escape genes that may favor sex-specific features; the involvement of differences in the expression of steroid hormones, and differences in anatomy, metabolism, gender or life experiences has also been hypothesized [25,26,27,28,29,30].

Interestingly, a higher incidence of several autoimmune diseases was detected in men with two X chromosomes (e.g., Klinefelter syndrome), whilst an opposite trend has been observed among females with a single X chromosome (Turner syndrome) [31,32,33], underlining the importance of the X chromosome in autoimmune disorders.

It is well known that several genes involved in immunity are located on the X chromosome (see below), as well as many genes involved in neurodevelopment or cancer. For instance, Dunford and colleagues found 6 out of 783 X chromosome genes with tumor-suppressive function had loss-of-function (LOF) mutations in males but not in females, extrapolating data from The Cancer Genome Atlas (TCGA) from 21 different tumor types [26]. The number of X chromosomes and the altered X-linked genes dosage seem, therefore, critical for the maintenance or the loss of the immune tolerance or for physiological processes, such as neurotransmitter biosynthesis and synaptic transmission, that is altered in neurological diseases.

On the other hand, sex hormones including estrogens, progesterone, androgens, and prolactin, can influence immune system function, predisposing the progression of autoimmune diseases.

They are known to act in a concentration-dependent manner, and their functions change based on the type of target cell and the receptor subtype expressed. Generally, 17-β estradiol (E2) and prolactin enhance humoral immunity, whilst testosterone and progesterone are known to be natural immunosuppressants decreasing pro-inflammatory mediators and inhibiting immune cell activation [16]. Estrogens also upregulate a wide range of immunity factors, including interferon (IFN) regulatory factor 5 (IRF5), and IFN-γ, as well as many others [16]. Additionally, post-menopausal women are also at increased risk of developing cardiovascular disease, which has been explained by a reduction in estrogen levels [34].

Estrogen protection regarding the female onset of hepatocellular carcinoma was observed, whilst the stimulatory effects of androgens in males increased both the frequency and aggressive phenotype [35]. Furthermore, the role of gonadal hormones in brain function was largely investigated as being causative of the female vulnerability to anxiety. In the same way, the changeable levels of hormones during women’s life, among perimenopause and menopause periods, as well as following giving birth, increase their susceptibility to depression and late-onset schizophrenia [8]. 

The protective properties of testosterone were observed in male animal models, in which anxiety behaviors resulted as reduced and the cognition was enhanced but was responsible for the severity of tics in males with Tourette’s syndrome [36,37,38]. Moreover, the neuroprotective action of estrogens was observed in neurodegenerative diseases, such as Parkinson’s disease, occurring early in men, that experience also distinct motor and nonmotor symptoms contrarily to Alzheimer’s disease, in which two-thirds of all patients are women [39].

It seems likely that gender-related biological differences arise during gestation; however, the contribution to human neurological disorders is not yet well understood, although it has been hypothesized that gender-specific events, including hormone signaling during early brain development could play relevant roles [7].

### 2.1. Differential Response to Infections

Certain viral and bacterial infections induce a differential response between males and females. Differences can regard the severity, prevalence, and pathogenesis of infections. For example, adult human females have higher antibody responses to influenza, hepatitis B, herpes virus, yellow fever, rabies, and smallpox virus vaccines than males [40].

Increased immunity to pathogens among females concurs to a lower intensity (i.e., viral load within an individual) and prevalence (i.e., the number of infected individuals within a population) of many infections among females compared to males, but may increase the symptoms and disease severity in females compared to males. For instance infections due to pathogenic influenza A viruses are often higher in men, but mortality is higher in women [41]. For other infections, the incidence is the same; however, the severity is higher in females (for example, measles, toxoplasmosis, dengue, and hantavirus [42]).

Males and females are believed to exhibit differential responses to infections mainly due to a different immune response pattern. The innate immune response, which is the first line of defense against any microbial infection, is based on the activity of pattern recognition receptors (PRRs), such as Toll-like receptors (TLRs) that specifically detect viral components, such as genomic DNA, double-stranded RNA, and single-stranded RNA, which regulate the production of type 1 interferon (IFN) as well as the production of inflammatory cytokines (TNF).

Studies in mouse and human models have shown that the number and activity of cells responsible for the innate immune response, such as monocytes, macrophages and dendritic cells, are higher in females than in males, in response to different antigens and different pathogens [43]. The adaptive immune response shows a sexually dimorphic response to infections. In general, females tend to show greater antibody responses over males, higher basal immunoglobulin levels, and higher B cell numbers [43,44].

Interestingly, a notable evolutionary benefit has been hypothesized represented by the fact that females of diverse vertebrate species have evolved to have increased antibody production over males, since the vertical transmission of maternal antibodies in utero or through breastfeeding represents a fundamental mean of protecting the offspring early in life [45,46].

Although gender differences in immune responses have been studied predominantly within the context of sex hormone effects, some other mechanisms for the hyper-responsiveness of the female immune system have been invoked: cellular mosaicism, genes escaping X chromosome inactivation, and some miRNAs encoded on the X chromosome that regulate the expression of molecules of the immune system (e.g., miR-18 and miR-19, which enhance inflammatory responses) [47]. Indeed, a higher innate resistance to infections was observed in females in the early stages of life, indicating that sex chromosomes, rather than hormonal status, play an important role in sexual differences in immunity [48]. 

The X chromosome is considered to have a fundamental role in determining sex-specific immune responses [47]. The X chromosome carries different genes involved in innate and adaptive immunity, among which genes involved in virus recognition (*TLR7* and *TLR8*); genes involved in the process of macrophage differentiation from hematopoietic stem cell (*IL3RA* and *GATA1*) and macrophage polarization (*IL13RA*); genes required for the activation of the intracellular oxidative burst in phagocytes (*CYBB*), a regulator of the actin cytoskeleton (*WAS*), and a gene involved in the TLR/IL-1R signaling pathways (*IRAK1*). Moreover, the X chromosome carries also different genes involved in adaptive immunity, such as *IL2RG*, *FOXP3*, and *CD40L* [47,48].

On the other hand, transcriptomic studies revealed a sex-biased expression in specific immune cell types, involving over 1875 genes. These transcriptional differences are highly cell-specific because most of these transcripts showed pronounced sex-biased expression patterns in only a single immune cell type. For example, in monocytes, a sex-biased expression was displayed by transcripts linked to the interferon pathway, including inflammatory cytokines and chemokines, like *IL6*, *TNF*, and *CXCL10*, and some females stood out for greater expression [49]. In this regard, it has been reported that, in women, an elevated induction of pro-inflammatory cytokines and chemokines correlated with higher morbidity and mortality due to influenza infections [50].

The importance of the mitochondria as regulators of the innate and adaptive immune response, in particular towards specific phenotypes of immune cells has been underlined by Kloc and coworkers, suggesting that the sex-related difference in the immune response may depend on sexually dimorphic populations of mitochondria. Similarly the so called “microgenderome”, the sex-dependent microbiome after puberty, which develops when the sex hormones start to work and regulates local and systemic inflammation, and response to infection [51], could have a role.

#### Response to Coronaviruses

Coronaviruses (CoVs) have been responsible of the severe acute respiratory syndrome (SARS)-CoV in 2002–2003 and of the Middle East respiratory syndrome (MERS)-CoV in 2012–2013 outbreaks.

Epidemiological data from those outbreaks indicated sex-dependent differences in disease outcomes. An in vivo study on a mouse model suggested that sex differences in the susceptibility to SARS-CoV in mice paralleled those observed in patients and also identified estrogen receptor signaling as critical for protection in females [52].

Recently, in December 2019, 27 patients from Wuhan, China, were found to be affected by viral pneumonia. The etiology of these infections was a novel coronavirus (2019-nCoV). Subsequently the International Committee on Taxonomy of Viruses renamed 2019-nCoV as “severe acute respiratory syndrome coronavirus-2” (SARS-CoV-2) [53].

Since then, a plethora of papers have been published on the pandemic of COVID-19, unfortunately, currently still ongoing, due to SARS-CoV-2. Among them, many clinical and epidemiological studies reported for the SARS-CoV-2 infection a sex-biased differential response, indicating higher morbidity and mortality in males over females. Data from China first revealed a gender bias in deaths, where 41.9% of the infected patients were women with greater lethality among males compared to females [54]. Subsequently also in other countries, a higher infection rate and severity of infection in males compared with females was confirmed [55]. However the emerging trend is that men and women have the same prevalence, but men with COVID-19 are more at risk for worse outcomes and death, independent of age [2,56,57,58]. On average, for each woman killed by COVID-19, 1.5 to 2 men succumb to the virus. This pattern is generally consistent around the world, including in the USA [59].

Gender differences in the response to inflammation have been documented and have been attributed, at least in part, to sex steroid hormones, as already reported. Specifically, the anti-inflammatory effects of estrogen and testosterone and the anabolic effect of testosterone, have been invoked. Moreover, age-associated decreases in estrogen and testosterone may mediate an increase in proinflammatory conditions in older adults that may enhance their risk of COVID-19 adverse outcomes [60].

One aspect that is more specifically related to the gender difference arises from the observation that, generally, women are more resistant to infections than men for several reasons, in addition to the abovementioned decrease in sex hormones and different immune response, due to their life style, as men typically smoke and drink more than women, while the latter have a greater aptitude with respect to men to take precautionary measures toward the COVID-19 pandemic, such as frequent hand washing, wearing of face mask, and stay at home orders [61].

### 2.2. Response to Vaccines

Several observational studies reported that the immune response to some vaccines differs between men and women. As mentioned above, adult females display stronger innate and adaptive immune responses than males, which can lead to a faster clearance of pathogens and greater vaccine efficacy in females than in males [42]. After vaccination against influenza, yellow fever, rubella, measles, mumps, hepatitis A and B, herpes simplex 2, rabies, smallpox, and dengue viruses, the protective antibody responses can be twice as high in females as males [62].

Indeed, in female mice vaccinated with inactivated H1N1 influenza, a greater antibody response was found, due to a greater TLR7 activation [44]. Moreover, females harbor more frequent and severe adverse reactions, such as fever, pain, and inflammation, to vaccines [62]. The explanation of the different response to vaccines between men and women, according to the literature is likely due to a number of determinants of gender differences, including age, reproductive status, hormonal differences, diet, microbiota, and genetic factors [40,63,64]. The biological differences between the sexes interfere with innate, adaptive, and memory immune responses, and can lead to adverse reactions (e.g., inflammatory responses) to vaccines, giving rise to gender differences in vaccine efficacy.

The stronger humoral response showed by women has been in particular related to high estrogen levels. Indeed, women appear to lose their immunological advantage after menopause [42]. Moreover, environmental exposures, including diet and coinfections, can interfere in particular with the microbiota, giving rise to differential responses to vaccines for males and females [40].

### 2.3. Gender-Specific Reactions to Xenobiotics

Several studies have shown gender differences in the response to a variety of environmental factors. Indeed, male and female organisms show significant differences in their toxicokinetics and response to chemicals [65,66].

Results from epidemiological and animal studies on the effects of lead (Pb) on children neurodevelopment indicated significant effect modification by sex of the brain function and developmental behavior. In particular, Pb neurotoxic effects appear to be more pronounced in males than females, although it should be considered that the influence of sex on outcomes depends on the type and the age when outcomes are measured [67]. Exposure to particulate matter, such as PM2.5 and black carbon, and to cigarette smoke seems to have gender-specific effects on health status, with men more sensitive to PM2.5 exposure, and women more susceptible to the harmful effects of cigarette smoke [68,69]. A recent revision of literature on the effects of caffeine consumption on the risk for neurological and psychiatric disorders showed that it can modulate occurrence of stroke, dementia, Parkinson’s disease, depression, and anxiety in a different manner in males and female [70].

Particularly, in the case of stroke, dementia, and depression, caffeine consumption seems to be more protective in women than in men, and has a more protective effect in men in the case of Parkinson’s disease [70]. There is also evidence of gender differences in the responses to treatments for several human pathologies, including cancer, cardiovascular disease, chronic pulmonary disease, stroke, diabetes, and migraine [20,71,72,73,74,75,76]. For example, in treatment of small cell-lung cancer, being female is generally regarded as a positive prognostic factor, although females are more prone to an increased toxicity from chemotherapy respect to males [77].

The prenatal period is a critical window for an organism’s development in which adverse experiences, including maternal stress and infections, are known risk factors for neurodevelopmental disorders, such as schizophrenia, autism, and attention deficit/hyperactivity disorder. Studies in rodent models demonstrated differences in the growth of the placenta, as well as altered adaptive responses to stressful environments between males and females [78]. For example, using mouse models of early prenatal stress, it has been observed that maternal stress increased the expression of immune response genes specifically in male placentas [79]. 

Moreover, in rats stressed prenatally, learning deficits and anxious behavior have been found to be more prevalent in males and females, respectively, although these effects depend also on the timing and intensity of the stress and the age when the offspring was tested [80]. Studies on rodents showed that even the exposure to endocrine disruptors, including bisphenol A (BPA) and phthalates, during critical periods of development is able to affect offspring health, with females more affected in the domain of emotionality, while males were more susceptible in the cognitive domain, specifically spatial memory [81]. Moreover, intrauterine exposure to phthalates has been associated with increased risk of growth retardation, a condition that predisposes to metabolic syndrome during adulthood, particularly in males [82].

## 3. Epigenetic Processes Involved in Gender Differences

Thus far, we have seen how the differences in the gender ratio occur for many complex traits and diseases. These different susceptibilities are believed to occur in many cases early in life, in accordance with the Developmental Origins of Health and Disease (DOHaD) theory. This theory postulates that environmental exposures in early childhood (pregnancy and first years of life: the first 1000 days), can alter the risk of disease throughout life until adulthood. 

Epigenetic modifications are likely the key effectors because they can be maintained through cell divisions. Epigenetic mechanisms, such as DNA methylation, histone tail modifications, and the non-coding RNA interventions, finely regulate the gene expression levels without inducing changes in the DNA sequence and have a fundamental role in embryonic development, differentiation, and the maintenance of cellular identity, as well as in many other physiological processes.

The two well-known cellular processes involving epigenetic mechanisms that primarily have a role in gender differences and which arise very early in embryo development are genomic imprinting and X chromosome inactivation in females.

### 3.1. Genomic Imprinting

Genomic imprinting is an epigenetic regulatory mechanism consisting of the monoallelic expression in the function of the parental origin of a subset of genes, located in specific regions, called differentially methylated regions (DMRs) [83]. These differential methylation sites located on the maternal and paternal alleles are protected from the wave of global demethylation that occurs immediately after fertilization, giving rise to a specific epigenetic signature of the parent of origin. The imprinting of these loci, that is the regulation of their gene expression, plays an essential role in the normal growth and development of placental mammals.

In humans, the loss of imprinting of specific DMRs results in a number of diseases often associated with impaired fetal growth and neurodevelopment, such as Beckwith-Wiedemann, Angelman, Prader-Willi, and Silver-Russell syndromes.

It is increasingly evident that the function of imprinted genes also plays a role in maternal physiology during reproduction. The imprinted genes are necessary for the development of a functional placenta, the organ that mediates the exchange of nutrients between mother and fetus. Since alteration in birth weight is related to adverse metabolic conditions in adults, including obesity and cardiovascular disease, modulation of this class of dose-sensitive and epigenetically regulated genes is likely to contribute to fetal and postnatal growth, with implications for health and disease throughout life [84].

It appears that aberrant imprinting can contribute to a variety of complex diseases. Many genome-wide studies found parental-origin-specific effects of genetic variants on complex traits/diseases, such as height, breast cancer, type 2 diabetes (T2D), and autism spectrum disorders, widening the functional role of imprinted genes in humans [85,86].

The phenomenon of genomic imprinting is mainly observed in eutherian mammals (mammals with long-lived placenta, which give birth to live young) including in marsupials, but not in prototherians (mammals that lay eggs), in birds or in reptiles [87]. Consistently, imprinting is observed to occur predominantly in the genes influencing fetal growth, particularly through placental growth, suckling, and nutrient metabolism [88].

#### 3.1.1. Placental-Specific Imprinting

The placenta is a temporary organ that regulates the growth and development of the embryo during gestation; it is mainly involved in the regulation of nutrient and gas exchange between the developing embryo and its mother. The extent of imprinted DMRs is extremely similar between tissues, with the exception of the placenta, in which several genes have been found to be specifically imprinted [89]. In fact, in the human placenta, there is a high retention of oocyte-derived methylation compared to the embryo, suggesting that epigenetic differences among the embryonic cells that establish extra-embryonic lineages at the time of implantation become fixed in the placenta [90,91]. The placenta may be able to uniquely maintain much maternal DMRs probably recruiting protective complexes [90].

Evidence has indicated the pivotal role of imprinted genes in fetoplacental development, regulating placenta implantation, growth, and embryogenesis [92]. Indeed, many imprinted genes have been associated with the fetal-growth promoting pathway, as well as the fetal-growth restricting pathways. The proper regulation of imprinted genes in the placenta is essential, while aberrant regulation is associated with abnormalities. In line with this, abnormal placental weights were observed in human infants with imprinting disorders, such as Beckwith–Wiedemann and Silver Russell syndromes [92]. 

Moreover, the aberrant expression of imprinted genes within placenta has been associated to altered fetal growth [93,94,95] and to infant neurobehavioral development in humans [96]; therefore, a correct expression of imprinted genes in the placenta is essential for fetal development. Moreover, the placenta imprinting status is very sensitive to early adverse environmental exposure. For example, altered placenta genomic imprinting has been observed in women exposed to endocrine disruptors, such as BPA and phthalates, as well as to residential air pollutants and alcohol consumption [97,98,99].

In addition to the placental-specific imprinting signature, the fetal sex can also modify the course and complications related to pregnancy and may also have an impact on the maternal health and well-being both during and after pregnancy. A review of literature on the effects of male and female fetal sex on the course of pregnancy and delivery, highlighted that the two sexes have different effects on pregnancy and delivery outcomes, as well as on maternal health [100]. Placenta function is also influenced by fetal sex, and placental sex seems to be a major determinant in the magnitude and functional responses of the placenta to perturbations during pregnancy [101]. 

For example, women who were pregnant with females were 2.55 at higher risk for placental malaria infection [102]. A recent meta-analysis of studies related to sex differences, prenatal exposures, and hypothalamic–pituitary–adrenal (HPA) axis reactivity found that female offspring exposed to stressors had increased HPA axis reactivity compared with males and that the female placenta increased its permeability to maternal glucocorticoids following maternal stress with changes in the expression of 11β-hydroxysteroid dehydrogenase enzymes in response to maternal glucocorticoid exposure or asthma [103].

Moreover, the increased placental oxidative/nitrative stress and reduced placental mitochondrial respiration related to altered maternal obesity and gestational diabetes mellitus is particularly evident when the fetus is male, suggesting a sexually dimorphic influence on the placenta [104]. Thus, it is very likely that a gender specific maternal-placental-fetal interaction exists, which influences the fetal and maternal biology through genetic and epigenetic mechanisms.

#### 3.1.2. Imprinting Deregulation, Brain Disorders, and Gender Differences

The discovery that many imprinted genes are more transcribed in the brain than in other somatic tissues has suggested their critical role in brain development and function. Many of these imprinted regions contain lncRNAs, which function as silencers of complementary sequences [105]. Early studies on imprinted genes indicated that some of them play a key role in determining brain size. Genes of maternal origin tend to favor brain growth while paternal alleles have the opposite action and limit brain growth. 

Moreover, several studies have shown that different cell types express specific imprinted genes and spatially localize to different regions of the developing brain [105]. Interestingly, a meta-analysis performed to search for differences in brain structure between males and females found regional sex differences in the volume and tissue density in the amygdala, hippocampus, and insula, which are areas known to be implicated in sex-biased neuropsychiatric and neurodegenerative conditions [106].

It is now widely accepted that imprinted genes are important regulators of variation in complex traits, and several imprinted genes have been involved in mental and physical development [107]. Human brain development requires an accurate orchestration of diverse spatial and temporal cues that modulate a regulatory interconnected network, and a failure in these pathways may lead to the development of cognitive or psychiatric disorders [108].

Although early studies on genomic imprinting highlighted its roles during embryonic and placental growth, its pleiotropic influences on neural development, wiring to synaptic function and plasticity, energy balance, social behaviors, emotions, and cognition are now emerging [109]. Different brain regions are maternally or paternally influenced during neurodevelopment, leading a gender-specific susceptibility to neurodevelopmental or behavioral disorders, including autism or schizophrenia [109,110]. The importance of the imprinted genes in brain function is also outlined by the devastating neurological and behavioral conditions that derive from mutations in imprinted loci [83,111,112].

### 3.2. The X-Chromosome Inactivation in Females

In female mammalian cells, one of the two X chromosomes is randomly inactivated early during embryogenesis in all cells, to balance the expression dosage between females and males, the latter having only one X chromosome. This X chromosome inactivation (XCI) represents a well-known epigenetic mechanism of gene regulation. Thus, females are mosaics for two cell populations: cells with the paternal X active and cells with the maternal X active. Since it is assumed that the choice of the X chromosome to inactivate is random, the result is that on average 50% of the cells express the paternal genes and the remaining 50% express the maternal genes. However, there are two phenomena in this regard that can interfere with what is expected: (1) in many cases, an unbalanced X inactivation has been recorded; (2) not all genes on the inactivated chromosome are silenced.

An asymmetric selection of the X chromosome to inactivate often occurs: it is known as skewed XCI (or non-random XCI). This is observed when more than 75–80% of the cells within a tissue inactivate the same X chromosome, while the extreme skewing of the XCI involves 90% of cell populations. Thus, skewed XCI could be detected in a tissue-specific manner or could affect the whole organism, depending on the gene-specific function and activity [113].

Skewed XCI can occur precociously by chance when the starting pool of progenitor cells is small, and the selection is preserved in the daughter cells. Ageing could also influence XCI promoting a skewed pathway in aged women following stochastic clonal loss or selection. Moreover, secondary stochastic events or genetic processes lead to selective pressure, as it was observed in adrenoleukodystrophy [114].

More generally, the onset of variants on the active X chromosome that potentially limit cell survival will undergo negative selection and exclusively cells carrying a viable gene propagate. XCI is incomplete: up to one-third of X-chromosomal genes are expressed from both the active and inactive X chromosomes (Xa and Xi, respectively) in female cells, with the degree of ‘escape’ from inactivation varying between genes and individuals [115].

Genes evading XCI have been detected particularly within pseudoautosomal regions (PARs), suggesting a “regionalization bias” (Figure 2). Escape genes within PAR1 region generally show male-bias in expression, while some PAR2 genes are silenced on both alleles [115,116,117,118].

A recent study, based on over 5500 transcriptomes from 449 individuals spanning 29 human tissues, provided an exhaustive survey of sex-biased gene expression in humans and demonstrated that expression of escape genes outside PARs is usually female-biased [115]. At least, 15–20% of X-linked genes outside of PARs escape silencing and show a gene-specific bias [118]. Moreover, most escape genes have shown a substantial heterogeneity across cell types, tissues, individuals, and experimental settings [115,117,119,121]. Tukiainen and co-workers showed that incomplete XCI affects at least 23% of X-chromosomal genes, and they identified seven genes that escape XCI. They suggested that escape from XCI results in sex differences in gene expression, thus, establishing incomplete XCI as a mechanism that is likely to introduce phenotypic diversity [115].

#### 3.2.1. Skewed XCI and Diseases

A growing body of evidence highlights a role of the skewed XCI in the female predisposition to several diseases. This can lead to hemizygosity of X-linked alleles that masks or unmasks X-linked variants that could be deleterious (e.g., certain recessive X-linked diseases that can develop also in women). Skewed XCI seems to be frequent in the healthy women population, it is not pathologic per se, and is also characterized by an age-dependent onset [122,123,124]. To date, the real prevalence of skewed XCI in unaffected population is still debated, probably due to differences in tissues examined, age of subjects, and methods used [115,122,125].

However, skewed X chromosome inactivation has been extensively investigated as a potential mechanism that contributes to the strong female preponderance observed in certain diseases. For instance, large studies investigated its role in the development of autoimmune diseases that show a strong female bias [126,127,128,129]. A skewed pattern of XCI has been detected in women affected by several diseases, including autoimmune thyroiditis, systemic sclerosis, systemic lupus erythematosus [130], scleroderma [131], and rheumatoid arthritis [127].

We already reported that several X-located genes are implicated in immunity (such as Toll-like receptors, cytokine receptors, genes related to T-cell and B-cell activity, and regulatory factors), and some of them are responsible for X-linked primary immunodeficiencies [116,117,118,119,120,121,122,123,124,125,126,127,128,129,130,131,132,133,134,135] (Figure 2).

Further, many genes associated with psychiatric and neurodegenerative diseases are located on the X chromosome, suggesting an involvement of the skewing XCI in the risk to the onset of these diseases. Indeed the effects of XCI have been implied in Rett syndrome, X-linked mental retardation, adrenoleukodystrophy, Aicardi syndrome, spinal and bulbar muscular atrophy, and Alzheimer’s disease [136,137,138,139].

In psychiatric disease studies, X-linked intellectual disability has been reported in association with skewed XCI involving variants in X chromosome genes, such as *MECP2*, *DDX3X*, and *SMC1A* [140]. A higher frequency of skewed XCI was also reported in children with autism (33%) than in age-matched healthy controls (11%) as well as in 50% of mothers with autistic daughters [141]. Additionally, skewed XCI in schizophrenia patients was found four-times more than that in the age- and sex-matched controls [142]. Finally, XCI skewing can have a role in cancer development; for example, expression of the breast tumor suppressor gene *FOXP3* was found altered by XCI skewing [143].

#### 3.2.2. Is There a Role for Escape Genes in Male-Female Difference?

As described above, XCI is the strategy adopted by females to balance the gene dosage between sexes. However, the mechanism of silencing is incomplete, resulting in 12% of genes that escape inactivation and another 15% of genes varying in their X chromosome inactivation status across individuals, tissues or cells: the expression levels of these genes are attributed to sex-dependent phenotypic variability [144,145,146]. Moreover, the sex chromosome complement was observed to influence the expression of autosomal genes, suggesting an additional global effect [147].

More evidence of the role of escape genes in sexual dimorphism has emerged from studies in human cancers. Several malignancies affecting the kidney and renal pelvis, blood, and brain cancers show a gender bias. Recently, it has been associated also with mutations in genes that escape XCI (“escape X chromosome loss” or “escape from X-inactivation tumour-suppressor (EXITS)” genes). Indeed, the X chromosome regions Xp11–22, Xq25–26, and Xq27–28 have been proposed as potential loci for tumor-suppressor genes that can escape from the X inactivation, including *ATRX*, *CNKSR2*, *DDX3X*, *KDM5C*, *KDM6A*, and *MAGEC3*, which were found to be more commonly mutated in males than females [26,148] (Figure 2).

These genes are required in one copy to prevent oncogenesis. Intuitively, males are consequently more vulnerable to mutations concerning females that result as protected through their Xi copy. Several of the immune-related genes on the X chromosome are known to escape and to be involved in the innate and adaptive immune response. In addition to the skewed XCI, the escape XCI may also provide a basis for more susceptibility of women to autoimmune diseases than men. Escape genes, including *CD40L*, *IRAK*-1, *TLR7*, *CXORF21,* and *XIAP*, are expressed in double dose compared to males, enhancing the susceptibility of women to autoimmune diseases, especially SLE. The role of all these genes is detailed reviewed in [145,149].

Furthermore, escape genes could be also important in brain function considering that 15% of the genes mapped on the X chromosome are involved in intellectual disability [148]. Somatic abnormalities that are eventually observed in females and males with sex chromosome aneuploidies (e.g., Turner or Klinefelter syndromes) could be explained with the escaping of genes involved in neurocognitive function, and female intellectual disability is a phenotype often associated to mutation on escape genes as well as to skewed XCI [149]. The involvement of escape genes covers a wide range of physiological and pathological mechanisms among females and among males and females that remain still to be fully explored.

## 4. Epigenetics as Unifying Mechanism

Many complex traits have a deviation of the gender ratio in humans. We have seen that epigenetic mechanisms are often involved in processes responsible of differential expression sex-related. We have also mentioned that, in the literature, there are several hypotheses to explain the deviation of the relationship between the sexes, mainly the hormonal and the genetic/chromosomal ones. Indeed, both can be interpreted in the light of epigenetic processes (Figure 3).

The close relationship between hormones and epigenetics is demonstrated by the fact that hormones can interfere with the transcription of target genes, in many ways determining gender-specific gene regulation. Many genes possess hormone response elements (HREs) in the promoter regions of their target genes (specifically, estrogen or androgen responsive elements are called EREs or AREs, respectively), which account for a direct effect of hormones on transcription [150,151]. Data obtained studying the effect of hormones on transcription of their target genes indicated that they can mediate active DNA demethylation occurring independently of DNA replication [152]. In the literature, some examples of molecular mechanisms involving methylation/demethylation by which hormones interfere with the transcription of target genes are known [152,153].

The expression of DNA methyltransferases (DNMTs) is under hormonal control. DNMTs are a family of enzymes that catalyze the reaction of addition of the methyl group to cytosine residues within CpG dinucleotides using S-adenosylmethionine as a methyl donor. A recent analysis of regulation of DNA methyltransferase isoforms in human breast tumors revealed that *DNMT1*, *DNMT3A,* and *DNMT3B* promoters harbor multiple ERα-binding sites [154].

A sex bias in DNA methylation levels, usually associated with sex-biased gene expression, was found in many animal and human studies [155,156,157]. In addition to gender effects for methylation loci on the X chromosome due to the X-inactivation dosage compensation mechanism in females, hundreds of autosomal CpG loci showed strong differences in the methylation between males and females [158,159]. This was confirmed recently by Zhuang et al., who studied the whole methylome and found a sex-biased DNA methylation in mouse and human livers [160].

Recent studies also explored another mechanism of gene regulation influenced by hormones: sex-specific chromatin accessibility. Chromatin remodeling, which is closely linked with accessibility to gene transcription, can be directly affected by members of the nuclear hormone receptor superfamily, including the steroid receptor (SR) subset containing the androgen receptor (AR) and estrogen receptor (ER). For instance, androgens can influence gene expression by decreasing the tri-methyl mark on lysine 27 of histone3 (H3K27me3), a gene silencing epigenetic mark, thus, determining sex- specific chromatin accessibility [161].

The evident change in prevalence around puberty for the frequency of asthma and allergic disease, which starts to change from being higher in males to higher in females, suggests that sex hormones and other factors alter pathways important in asthma pathogenesis and allergic disease [162]. On the other hand, allergic diseases can be considered prototypic examples of conditions determined by GxE (gene × environment) interactions [163].

In this context, many studies have shown that environmental exposures, such as microbial and dietary, but especially air pollution, increase the risk of developing allergic disorders, and these effects are related to the altered methylome profile in exposed subjects affected by allergies [163]. Moreover, in boys and girls affected by childhood asthma, sex specific DNA methylation changes were found at the promoter region of the *ZPBP2* gene, with lower methylation levels in boys compared with girls. DNA methylation also varied with age and was higher in adult males compared to boys, suggesting that this epigenetic mechanism could mediate the sex and age specific associations [164].

In the 1980s, an imprinting process called “hormonal imprinting” was described by Csaba [165]. It refers to a physiological process in which the developing hormone receptor encounters the target hormone for the first time and consequently the receptor-hormone pair is established, and this effect lasts for life. However, during embryogenesis, in particular in the phases of endocrine system development, receptors do not show complete specificity, consequently molecules similar to the target hormone (such as synthetic molecules, hormone analogues, endocrine disruptors) are also able to be tied up by the receptors, causing defective hormonal imprinting, with potential lifelong consequences [166].

According to Csaba (2019), this process can be viewed as an epigenetic process. Faulty hormonal imprinting is a disturbance of gene methylation pattern, which is epigenetically inherited to the further generations (transgenerational imprinting), as it would imply an epigenetic reprogramming of both receptor cells and germ cells, with transgenerational transmission. The effect of transgenerational transmission was observed in the third generation in case of prenatal exposure to diethylstilbestrol (DES), in humans.

DES third-generation women may have an increased risk of irregular menstrual cycles, amenorrhea, and preterm delivery; in addition they rarely develop a specific form of cancer: clear-cell adenocarcinoma of the vagina. These findings are considered consistent with an intergenerational transmission of epigenetic alterations affecting the primordial germ cells of the DES-exposed fetus [167]. This can be manifested in altered binding capacity of the receptor, with the disturbed transmission of the message (contained in the hormone-like molecule) and the disturbed response of the receptor-bearing cell [166].

This mechanism could be of primary importance in the frame of the DOHaD theory and for the susceptibility to age-related diseases. Gender-specific differences in epigenetic marks could influence the differential susceptibility to neurological disorders, as several sex-associated differentially methylated regions were identified between females and males [168,169].

It should also be outlined that the majority of the Mendelian diseases of the epigenetic machinery, due to mutations in genes encoding for the enzymes that write, read, erase, or remodel epigenetic marks, are characterized by neurological dysfunction, particularly intellectual disability [170]. Therefore, the brain is emerging as an important target of genomic imprinting and epigenetic regulation, focusing attention on how these processes influence neural development and behavior. Neurodevelopmental disorders may originate from a failure in the epigenetic signaling during fetal or perinatal brain development, which may affect males and females differently.

Studies performed in animal models and in human birth cohorts have revealed that the same developmental periods in which epigenetic programming is more prone to undergo alterations, resulted also sensitive to environmental insults, suggesting that epigenetics represent a potential mechanism through which sexually dimorphic effects of early-life exposures could manifest [171].

For instance, animal and human studies showed that perinatal exposure to Pb resulted in sex dependent DNA methylation alterations in the adult [172,173,174,175,176,177]. It has been reported that early childhood Pb exposure resulted in peripheral blood leukocytes sex-dependent DNA methylation differences in the DMRs of the imprinted regions *PEG3*, *IGF2*/*H19*, and *PLAGL1*/*HYMAI* in adulthood [176]. Either exposure to endocrine disrupting chemicals, including BPA and phthalates, were found to induce altered methylation levels in infants cord blood and peripheral blood of several genes, such as *IGF2*, *H19* and *PPARA* and of LINE-1 sequences, in a gender-specific manner [178,179].

The chromosome theory tries to explain the gender bias present in many complex traits clinging to differences in gene content, genetic variants, and gene expression dosage between the X and Y chromosomes. Actually, as shown above, the X-inactivation process in females (which in some circumstances can be skewed) in relation to the presence of many genes involved in immunity, the dosage imbalance generated by incomplete X chromosome inactivation (involving escape genes) can justify the difference, in terms of differential gene expression, that means epigenetic regulation.

It has also been hypothesized that sex differences due to X chromosome-associated mechanisms, derive by differential expression of X-linked microRNAs (miRNAs). These small non-coding RNAs (21–25 nucleotides in length) are epigenetic effectors as they are involved in post-transcriptional gene expression regulation, by carrying on translational repression and/or messenger RNA degradation. Their regulatory power is well recognized, as 30–50% of all protein-coding genes are targeted by miRNAs, each gene might be regulated by many miRNAs and each single miRNA targets hundreds of genes [73].

The human X chromosome contains 10% of all microRNAs detected so far in the human genome, and particularly, it contains a higher number of microRNAs, at present 118, when compared to those contained in the Y chromosome (two miRNAs), and to those contained in autosomes (an average of 40–50 miRNAs) [73]. The function of the majority of the X chromosome encoded miRNAs remain to be elucidated, although it is known that most of them are involved in cell lineage determination, in cytokine production and in apoptosis regulation [119]. Potential biological targets of X-linked miRNAs include genes that play an important role in the immune suppressive mechanisms and maintenance of tolerance [180].

It is, therefore, not surprising that altered expression of the X-chromosome-linked miRNAs have been found in human tissues of patients affected by several diseases, including autoimmune diseases and cancer. For example, expression of mir-18b (gene location Xq26.2), has been found altered in peripheral blood of multiple sclerosis patients [181], in breast cancer tissue [182], in primary melanoma cells [183] and in hepatocellular carcinoma tissue [184]. Since several miRNAs on X-chromosome are located in genes that have been shown to escape XCI, such as *DMD*, *CHM*, *ATP11C*, or *IRAK* (Figure 2), disturbances in the normal inactivation pattern of the X chromosome-associated miRNAs, could affect miRNAs-driven gene regulation and result in sex-specific responses [119].

The differential expression of X-chromosome miRNAs between male and females have been suggested to underlie differences between the two sexes in response to cerebral ischemia [74], sex disparity in cancer [73] and heart failure [185]. However, more evidence of variation of X-linked miRNAs according to sex chromosomes, has been obtained regarding the different impact among sexes of autoimmune diseases development [186]. For example, X-linked miRNAs, including miR-188, miR-421, miR-503, let-7f-2, and miR-98, were found to be upregulated in CD4+ T cells from female lupus patients compared to male lupus patients potentially contributing to the sex difference in lupus [187].

It should be hypothesized that female mosaicism, silencing escape or skewed patterns of inactivation of X-linked miRNAs involved in immunity could lead to unbalanced miRNAs expression between sexes, and to sex-specific immune responses [186]. It has already been mentioned that some of the genes differentially expressed between men and women involved in innate and adaptive immunity and located on the X chromosome, are important epigenetic regulators. Another gene implicated in immune responses against viral infections and in autoimmune diseases is *KDM6A*, coding an epigenetic regulator (lysine-specific demethylase 6A). It has been hypothesized that this may play a role in immune responses against COVID-19 [188].

Viral infections establish a network of virus-host functional interactions, which mainly antagonize the regulatory machine of the host by altering its metabolism and gene expression, setting up a permissive environment for virus replication and spread which depends on epigenetic modifications, mainly chromatin dynamic [189]). In particular, some studies suggest that the effects of respiratory viral infections are at least in part mediated by epigenetic changes (such as altered levels of DNA methylation or dysregulation of specific miRNAs) in airway epithelial cells [190]. Coronaviruses, such as MERS-CoV and SARS-CoV-1, are known to mediate epigenetic alterations by antagonizing host antigen presentation or activating interferon-response genes [191].

Regarding specifically SARS-CoV-2 infection, recent findings have revealed that global DNA methylation and specific *ACE2* gene methylation as well as post-translational histone modifications may drive differences in host tissue-, biological age- as well as sex-biased patterns of viral infection [192]. ACE2, the angiotensin-converting enzyme 2, identified as a crucial factor that facilitates SARS-CoV-2 to bind and enter host cells, was found among the genes that escape X inactivation in female tissue, showing heterogeneous sex bias in various tissues [115].

Moreover, *ACE2* expression was a little higher in male than in female tissues specifically in lugs of male patients with comorbidities associated with severe COVID-19, compared to control individuals [193]. Interestingly *ACE2* lies in Xq, within the evolutionarily older region of the chromosome, where escape genes also show higher tissue-specificity and lower expression levels [194]. Epigenetic control of the *ACE2* gene might be a target for prevention and therapy in COVID-19 [195]. Finally, the same considerations made about the greater reactivity of women to viral infections, which is expressed through epigenetic mechanisms of gene regulation (see Figure 3), may also be valid for explaining the greater female reactivity to vaccines, as well as different outcomes of vaccine safety and efficacy.

## 5. Concluding Remarks

An increasing number of literature studies have reported that epigenetic dysregulation plays an important role in complex human traits/diseases and, in a considerable part of them, gender differences have been reported. Certainly, there are many factors involved in gender differences.

A significant proportion of human imprinted genes are realistically involved, as well as altered expression of critical regions, due for instance to a skewed inactivation of the X-chromosome in females, or to the presence of genes that escape to X-inactivation or to a deregulation of miRNAs located on the X chromosome. Moreover, the expression of specific genes can be modified epigenetically by many intracellular (hormones) or extracellular environmental factors. Likely, all these players act in the most vulnerable stages of our life, such as pregnancy, by interfering with fetal programming and making individuals more susceptible to developing diseases later in life.

The need to including a gender dimension in clinical studies and practice is increasingly felt, especially with the current growing interest in precision medicine. Drug therapies are not yet optimized for both males and females, although women are now included in clinical drug trials.

The different efficacy of drugs in women and men is due to biological differences that may be caused by sex-specific gene expression, which is likely triggered by sex-specific epigenetic modifications. Furthermore, gender plays a role in drug efficacy as a sociocultural dimension that can lead to differences between women and men. The same considerations can also be valid for vaccines. Many studies clearly demonstrate that the development of vaccines against infections should carefully consider the effect of differential immune response in males and females [40]. Moreover, also an accurate analysis of outcomes as well as of adverse effects should be performed.

Even if, to date, no extensive studies have been published on the possible gender bias regarding the efficacy and any adverse reactions to vaccines against SARS-CoV-2 infection, some recent reports indicate that certain vaccines (ChAdOx1 nCoV-19 and the Ad26.COV2. S, which are both DNA-vectored vaccines against SARS-CoV-2 infection) can induce a rare clinical syndrome characterized by thrombosis at atypical sites combined with thrombocytopenia, occurring within 2 weeks of vaccination in women under 60 years of age, with lethality due to thrombosis in a minority of them [196,197,198].

It seems in agreement with what was said above that females typically produce higher antibody responses and show more adverse effects of vaccination respect to males. In particular it may be due the production of autoantibodies against the Platelet Factor-4 (PF4) in individuals susceptible to autoimmune diseases [199]. We can speculate that the involvement of young women in particular suggests a reaction involving the immune system, which becomes less severe in older women.

Therefore, it would be of fundamental importance to take these gender-related aspects into consideration also in the frame of the huge vaccination campaign against SARS-Cov-2 still in progress. The proposal by Flanaghan and coworkers is at present particularly valuable “all vaccine studies should consider sex and gender across the life span in the design of vaccines and analysis of outcome measures” [40].

## Figures and Tables

**Figure 1 biomedicines-09-00652-f001:**
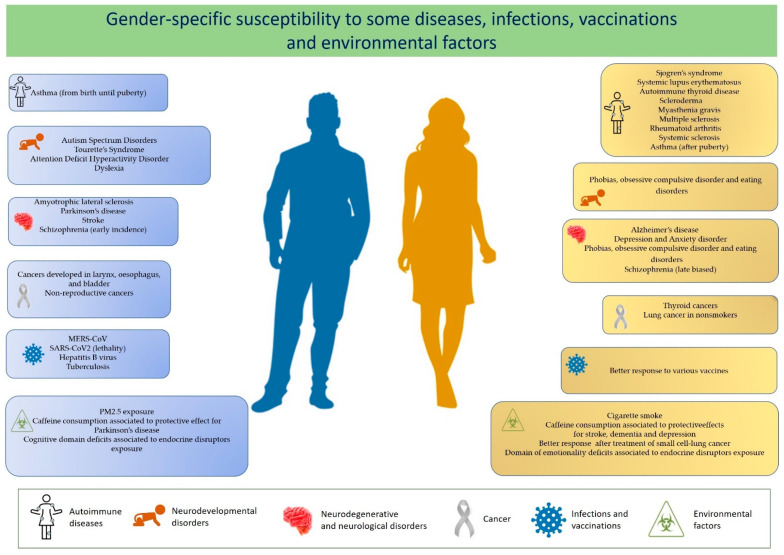
Increased susceptibility to some diseases and infections and to the effects of vaccinations and environmental factors in males (left panels) and females (right panels).

**Figure 2 biomedicines-09-00652-f002:**
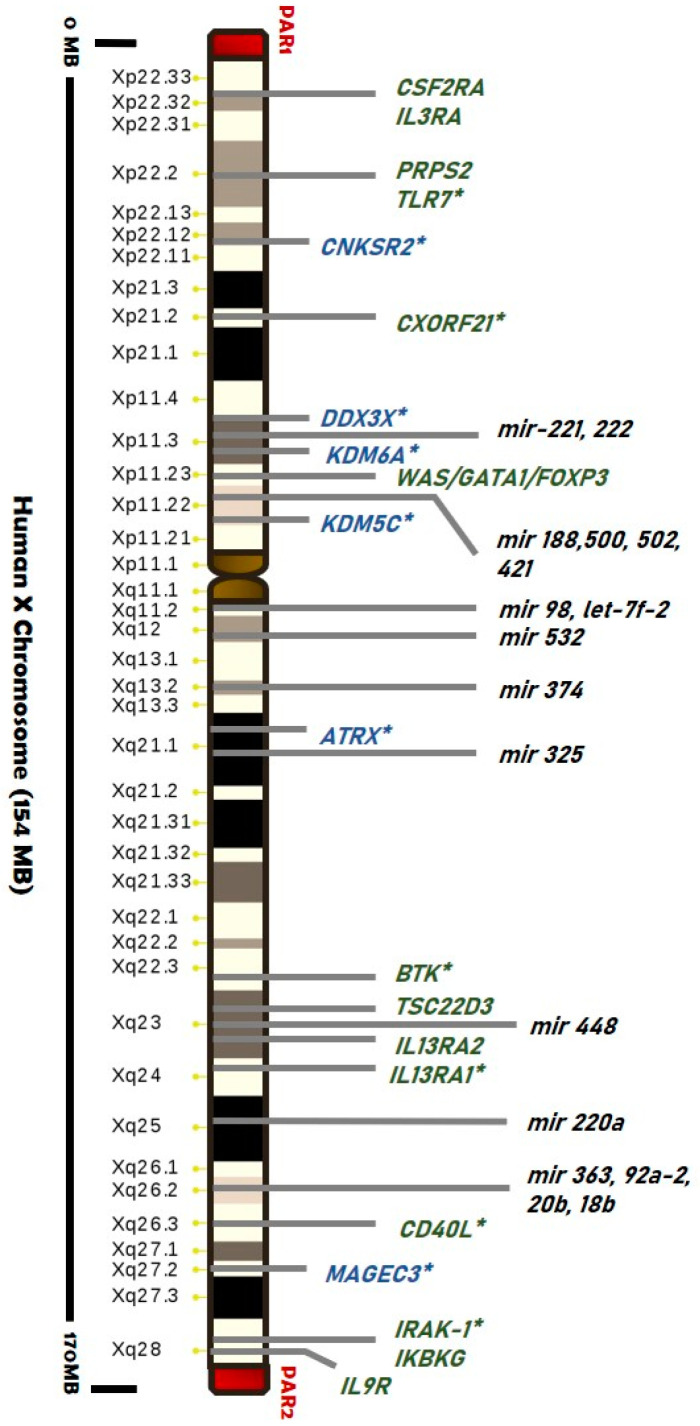
Map of the main genes and miRNAs (black) located on the human X chromosome associated with cancer development (blue) and autoimmunity (green). Genes that escape from the X-inactivation are labeled (*) (list of genes from [119,120]). Pseudoautosomal Region 1 and 2 (red blocks) and centromere (brown block) are shown.

**Figure 3 biomedicines-09-00652-f003:**
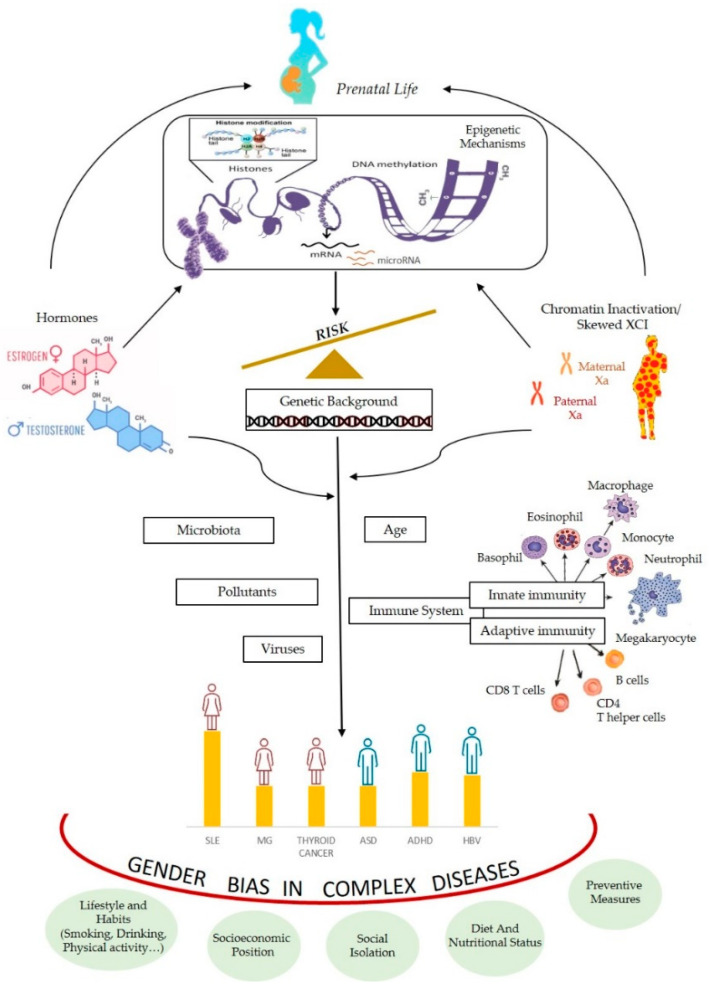
An integrated model of the multiple factors interfering with gender bias in complex diseases throughout epigenetic mechanisms (Xa; active X chromosome).

**Table 1 biomedicines-09-00652-t001:** Examples of the female:male ratios for diseases.

	Female to Male Ratio	References
**Autoimmune disorders**		
Sjogren’s syndrome (SS)	9:1	[5]
Systemic lupus erythematosus (SLE)	7:1	[5]
Autoimmune thyroid disease (AITD)	7:1–10:1	[6]
Scleroderma	7:1	[6]
Myasthenia gravis (MG)	2:1–3:1	[6]
Rheumatoid arthritis (RA)	2–3:1	[5,6]
Multiple sclerosis	2–3:1	[7]
Systemic sclerosis	3:1	[6]
**Neurodevelopmental and Neurodegenerative diseases**		
Autism Spectrum Disorders (ASD)	1:3–4 1:9 (High-functioning patients)	[7,8]
Attention Deficit Hyperactivity Disorder (ADHD)	1:3	[8]
Tourette’s Syndrome	1:4	[8]
Depression and Anxiety disorder	2:1	[7,8]
Amyotrophic lateral sclerosis	1:1.6	[7]
Schizophrenia	Early incidence male-biased Late incidence female-biased	[7]
Stroke	1:2	[7]
Parkinson’s disease	1:1.5	[8,9]
Alzheimer’s disease	1.6–3:1	[7]
**Cancer**		
Cancers developed in larynx, esophagus, and bladder	1:4	[10]
Non-reproductive cancers	1:2	[10]
Thyroid cancers	3:1	[11]
**Infectious diseases**		
MERS-CoV	1:2	[12]
SARS-CoV2 (lethality)	1:1.5	[13]
Hepatitis B virus	1:3.8	[14]
Tuberculosis	1:1.6	[15]

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
