# Peer review of "Gender Specific Differences in Disease Susceptibility: The Role of Epigenetics"

_biomedicines, 2021, doi:10.3390/biomedicines9060652_

Round 1
Reviewer 1 Report
Manuscript by Migliore et al., has very nicely reviewed the literature available for Gender specific differences in disease susceptibility and highlighted the role of epigenetics.
Comment: 1. As authors have discussed about the COVID viruses. However, the gender bias in response to vaccination is not very well explained. Authors are suggested to incorporate current vaccination data across different countries for recent COVID and see if there is any gender bias here in response to the vaccines, recurrence of disease etc. Also, there were reports where women died from complications after receiving Johnson & Johnson vaccines. What do authors think about such data? Is there a bias here, please discuss.Author Response
Referee 1
Comment: 1. As authors have discussed about the COVID viruses. However, the gender bias in response to vaccination is not very well explained. Authors are suggested to incorporate current vaccination data across different countries for recent COVID and see if there is any gender bias here in response to the vaccines, recurrence of disease etc. Also, there were reports where women died from complications after receiving Johnson & Johnson vaccines. What do authors think about such data? Is there a bias here, please discuss.
Answer to Referee 1
We would like to thank the Referee who made useful and justified comments, who helped us to improve the manuscript.
The topic relating to vaccination has been somewhat expanded, to answer the question posed by the referee. To avoid placing too much emphasis on COVID-19, as Referee 2 argues, we slightly expanded the part on vaccines in the conclusions (page 17, lines 1192-1204).
Reviewer 2 Report
With interest, I read the manuscript biomedicines-1224884.
Comments:
- There is lots of information in this article, however, only some parts of it are summarized (two tables), not necessarily those most important. The Authors should add 3-4 comprehensive figures outlining the major contents of the article, especially those more directly related to epigenetics.
- While discussing the examples of sex differences in the epigenetic background of diseases, please, refer to allergies as well, as outlined in the respective section of PMID: 28322581. Allergies represent a prototypic gene x environment (epigenetics) disease.
- I understand that COVID-19 is now trendy. More importantly, I can see that you gather lots of data on this infectious disorder. But you need to make clear how you got there. I mean, you topic is very wide so you need to make some selection but the article needs some flow, so you need to integrate this COVID-19 better. At the moment, it is a bit a foreign body.
- While talking on the epigenetic mechanisms contributing to COVID-19, please, shortly mention the involvement of epigenetic mechanisms in infections with some other respiratory viruses. Please, refer to the respective section of PMID: 32973742.
Other comments:
- Line 535. “4. Epigenetic As Unifying Mechanism” -> “4. Epigenetics As Unifying Mechanism”.
Author Response
Referee 2
Comments:
- There is lots of information in this article, however, only some parts of it are summarized (two tables), not necessarily those most important. The Authors should add 3-4 comprehensive figures outlining the major contents of the article, especially those more directly related to epigenetics.
- While discussing the examples of sex differences in the epigenetic background of diseases, please, refer to allergies as well, as outlined in the respective section of PMID: 28322581. Allergies represent a prototypic gene x environment (epigenetics) disease.
- I understand that COVID-19 is now trendy. More importantly, I can see that you gather lots of data on this infectious disorder. But you need to make clear how you got there. I mean, you topic is very wide so you need to make some selection but the article needs some flow, so you need to integrate this COVID-19 better. At the moment, it is a bit a foreign body.
- While talking on the epigenetic mechanisms contributing to COVID-19, please, shortly mention the involvement of epigenetic mechanisms in infections with some other respiratory viruses. Please, refer to the respective section of PMID: 32973742.
Other comments:
- Line 535. “4. Epigeneti As Unifying Mechanism” -> “4. Epigenetics As Unifying Mechanism”.
Answers to Referee 2
We would like to thank the Referee who made useful and justified comments, who helped us to improve the manuscript.
- The first item has been addressed improving the iconographic aspect (tables and figures) according to the Referee’s suggestions:
Table 2 has been deleted and 3 figures have been added to better graphically represent the concepts described in the various paragraphs, in particular in paragraph 4.
- We agree with the Referee's comment that allergies are an excellent example of gene-environment interaction, with epigenetic implications, not previously included by us.
Allergies have now been addressed (see page 4, lines 98-101; page 15, lines 967-979).
- To address this point we have decided to eliminate Tab.2, leaving in the text the references to COVID-19 which we consider appropriate and better integrated (see page 6-7, lines 302-396; page 17, lines 1138-1150)
- As mentioned above, a better integration of Sars-Cov 2 was also obtained by inserting, as suggested by Referee 2, epigenetic mechanisms in infections with some other respiratory viruses (see page 17, lines 1130-1138).
Line 535. “4. Epigeneti As Unifying Mechanism” has been corrected in “4. Epigenetics As Unifying Mechanism”.
Round 2
Reviewer 2 Report
Thank you very much for adhering to my suggestions. The figures are excellent.